# Mapping Annual Tidal Flat Loss and Gain in the Micro-Tidal Area Integrating Dual Full-Time Series Spectral Indices

Jiayi Luo [1,2,†], Wenting Cao [2,†], Xuecao Li [2,3], Yuyu Zhou [4], Shuangyan He [1], Zhaoyuan Zhang [2], Dongling Li [2] and Huaguo Zhang [1,2,5,*]

1   Ocean College, Zhejiang University, Zhoushan 316021, China; 22134137@zju.edu.cn (J.L.);
    hesy103@163.com (S.H.)
2   State Key Laboratory of Satellite Ocean Environment Dynamics, Second Institute of Oceanography,
    Ministry of Natural Resources, Hangzhou 310012, China; caowt@sio.org.cn (W.C.);
    xuecaoli@cau.edu.cn (X.L.); zhang_zy@sjtu.edu.cn (Z.Z.); ldl@sio.org.cn (D.L.)
3   College of Land Science and Technology, China Agricultural University, Beijing 100083, China
4   Department of Geography, Institute for Climate and Carbon Neutrality, The University of Hong Kong,
    Hong Kong, China; yuyuzhou@hku.hk
5   Observation and Research Station for Marine Risk and Hazard Management at Daya Bay,
    Ministry of Natural Resources, Huizhou 516081, China
*   Correspondence: zhanghg@sio.org.cn
†   These authors contributed equally to this work.

**Abstract:** Tracking long-term tidal flat dynamics is crucial for coastal restoration decision making. Accurately capturing the loss and gain of tidal flats due to human-induced disturbances is challenging in the micro-tidal areas. In this study, we developed an automated method for mapping the annual tidal flat changes in the micro-tidal areas under intense human activities, by integrating spectral harmonization, time series segmentation from dual spectral indices, and the tide-independent hierarchical classification strategy. Our method has two key novelties. First, we adopt flexible temporal segments for each pixel based on the dual full-time series spectral indices, instead of solely using a fixed period window, to help obtain more reliable inundation frequency features. Second, a tide-independent hierarchical classification strategy based on the inundation features and the Otsu algorithm capture the tidal flat changes well. Our method performed well in Guangdong, Hong Kong, and Macao (GHKM), a typical area with micro-tidal range and intense human activities, with overall accuracies of 89% and 92% for conversion types and turning years, respectively. The tidal flats in GHKM decreased by 24% from 1986 to 2021, resulting from the loss of 504.45 km$^2$, partially offset by an accretion of 179.88 km$^2$. Further, 70.9% of the total loss was in the Great Bay Area, concentrated in 1991–1998 and 2001–2016. The historical trajectories of tidal flat loss were driven by various policies implemented by the national, provincial, and local governments. Our method is promising for extension to other micro-tidal areas to provide more scientific support for coastal resource management and restoration.

**Keywords:** full-time series indices; annual tidal flats; change detection; spectral harmonization; land reclamation

## 1. Introduction

Tidal flats, located between the mean high water and the mean low water, are periodically exposed areas without vegetation [1]. Tidal flats play essential roles in protecting the ecological environment as well as providing food and habitats [2–6]. However, with the intensification of coastal human activities, tidal flat resources are being reclaimed for various purposes including agriculture, industry, residential areas, recreation, and commercial development [7,8]. Such rapid urbanization and economic development have significantly decreased the tidal flats around the world [7,9–12]. Therefore, there is an urgent need to enhance our understanding of the long-term loss and gain dynamics in tidal flats to provide scientific support for coastal restoration decision making.

The advancements in remote sensing technology have effectively reduced the human resources, materials, and economic expenses in mapping tidal flats compared with field surveys [13–15]. The traditional remote sensing methods rely on cloud-free satellite images and employ visual interpretation, supervised classification, or the waterline extraction approach to delineate the tidal flat areas between the instantaneous waterlines at high and low tides [16–19]. However, due to the relatively fixed imaging time of remote sensing satellites, fluctuations in the tidal heights, and the frequent cloudy and rainy weather in coastal areas, the availability of remote sensing images with cloud-free coverage at high or low tides is limited [12,20]. As a result, these traditional methods require manual screening of numerous images to identify the limited cloud-free images captured at high or low tides. This limits the mapping results to sparse multi-temporal changes, leading to difficulties in achieving high spatiotemporal continuous monitoring of tidal flat changes.

To address this issue, recent studies have attempted to use time series remote sensing images to map tidal flats, without the dependence on cloud-free images at high or low tides [21–26]. Fully utilizing all the cloud-free pixels from time series remote sensing images can provide fruitful spectra temporal features and capture more periodic inundation dynamics of tidal flats at the pixel level. Specifically, a few studies used Landsat time series spectral stacks and machine learning classification methods for mapping tidal flat changes with reliable performance [14,19,27,28]. Nevertheless, such supervised classification methods rely on a substantial amount of high-precision training samples, leading to challenges in real-time monitoring updates in areas with significant changes. To improve the timeliness and automation of mapping tidal flat changes, recent studies have used time series remote sensing to calculate the occurrence of inundation, without relying on training samples. Representative studies such as Wang et al. [12] and Chen et al. [21] mapped the long-term tidal flats by calculating the inundation frequency during fixed natural periods (1-year and 3-year, respectively), and found that the Yellow River and Yangtze River Delta in China were the regions with significant tidal flat loss. But calculating inundation frequency within fixed periods may introduce bias in the frequency values due to factors such as insufficient observation data and coastal land cover changes within the fixed periods, thus limiting the accurate mapping of tidal flats. Additionally, Cao et al. [8] used flexible temporal segments of varying lengths for each pixel to calculate the inundation frequency, and effectively captured tidal flat loss caused by land reclamation in the macro-tidal areas with tidal ranges of 6–8 m.

In summary, the inundation frequency-based tidal flat mapping method has the advantages of not relying on training samples, being automated, and applicable to large-scale areas. However, time series spectral characteristics used to represent inundation status are influenced by regional environments and their ability to map the tidal flats in the micro-tidal area remains unclear. Existing methods have paid more attention to macro-tidal or meso-tidal range areas, while limited attention given to the micro-tidal areas. Micro-tide refers to a tidal range of less than 2 m, while meso-tide refers to a range between 2 and 4 m, and macro-tide refers to a range larger than 4 m [29]. Due to the small tidal range, the mapping of tidal flats in the micro-tidal area is more challenging compared with other areas. Specifically, a large tidal range is particularly conducive to the formation of flats, so the tidal flat area in the macro-tidal and meso-tidal area is generally relatively extensive [30]. In contrast, tidal flats in micro-tidal area usually have a low elevation and high water content, leading to a greater likelihood of spectra confusion between the tidal flats and water, especially along the seaward boundary of the tidal flats. In this context, accurately capturing the loss and gain of tidal flats due to human-induced disturbances becomes more challenging. More efforts should be made to investigate the time series spectra for capturing the tidal flat loss and gain in micro-tidal areas under significant human activities.

Guangdong, Hong Kong, and Macao (GHKM) is a typical area with micro-tidal range and intense human activities. Specifically, tidal flats in GHKM are widely distributed along approximately 4500 km of coastline in the South China Sea where the tidal range is lower than 2 m [31], and they provide numerous ecosystem services for more than 134 million

populations. Meanwhile, as one of the most populous and developed urban agglomerations in the world, GHKM's unprecedented urbanization progress has squeezed the extent of tidal flats. Recently, GHKM has been one of the key areas of the national coastal protection and restoration projects, according to the China's master plan (2021–2035) for major projects to protect and restore key ecosystems. However, existing studies in GHKM primarily focus on mangroves or urban land cover changes [25,32–35], with relatively limited attention on the tidal flats. Therefore, it is essential to capturing the long-term annual loss and gain of tidal flats in GHKM to provide a scientific basis for the national coastal protection and restoration projects.

In this study, taking GHKM as the study area, we performed spectral consistency harmonization for different sensors, screened dual spectral indices by comparing the capabilities of four commonly used remote sensing indices, thereby developing an automated method for mapping the annual tidal flat changes by integrating time series segmentation and hierarchical classification strategy. We applied the method and obtained the long-term annual tidal flat loss and gain dynamics in the GHKM during 1986–2021.

## 2. The Study Area and the Dataset

### 2.1. The Study Area

GHKM is located in South China, composed of the Guangdong province and two special administrative regions, Hong Kong and Macao (Figure 1a). It encompasses both tropical and subtropical zones, and experiences hot and rainy summers, as well as warm and humid winters [36]. The tides in GHKM are a mixed tide type of semi-diurnal tide and diurnal tide, with the diurnal tide being the main one. By 2021, the total population of GHKM reached 134.9 million, accounting for 11.3% of China's total population, and the GDP exceeded 12 trillion yuan, accounting for 10% of the national GDP. From 1986 to 2021, the GDP in GHKM increased by 22,652 billion dollars at an annual growth rate of 10.9%. Similarly, the total population in GHKM has increased by 71 million at an annual growth rate of 2.2% [37]. The coastal zone of GHKM has three sub-regions, i.e., the Western region, the Greater Bay Area (GBA), and the Eastern region, according to their geographical location and economic development level [38,39] (Figure 1a). The GDP of the Western region, the GBA, and the Eastern region accounted for 7.0%, 80.9%, and 6.2% of the GHKM's total GDP in 2021, respectively [40].

GHKM has abundant tidal flat resources along the coasts which have been undergoing significant reclamation due to the rapid economic development since 1978 [25,36]. Recently, GHKM has been listed as the key area of the China's master plan (2021–2035) for major projects to protect and restore key ecosystems. Thus, it is essential to advance our understanding of long-term tidal flat loss and gain in GHKM. This will provide vital scientific support for decision making in coastal restoration efforts. In this study, we established a 2 km seaward buffer zone using the coastline of 1986 and make sure that it covers all the tidal flat extents in 1986 and 2021 by manual adjustments, thereby defining the study area with a total area of 2946.01 km$^2$.

### 2.2. The Dataset

All the available Landsat surface reflectance Tier 1 dataset covering the study area in Google Earth Engine (GEE) from 1986 to 2021 was adopted in this study, with 6958 scenes in total. The dataset with a spatial resolution of 30 m has been geometrically corrected using the LPGS code and converted into atmospherically corrected surface reflectance [41,42]. Specifically, the dataset was atmospherically corrected using the Landsat Ecosystem Disturbance Adaptive Processing System (LEDAPS) algorithm for the Thematic Mapper (TM) and Enhanced Thematic Mapper Plus (ETM+) sensors and the LaSRC algorithm for the Operational Land Imager (OLI) sensor [43]. The subsequent analyses used all the available pixels after cloud and cloud shadow masking [44] (Figure 1b). The Landsat data used in this study consisted of 2976 TM images from 1986 to 2011, 2599 ETM+ images from 1999 to 2021, and 1383 OLI images from 2013 to 2021 (Figure 1c). Notably, most of the images

covering the GHKM had a cloud cover of less than 80% (Figure 1d). More details about the Landsat time series data preprocessing could be found in Section 3.1.

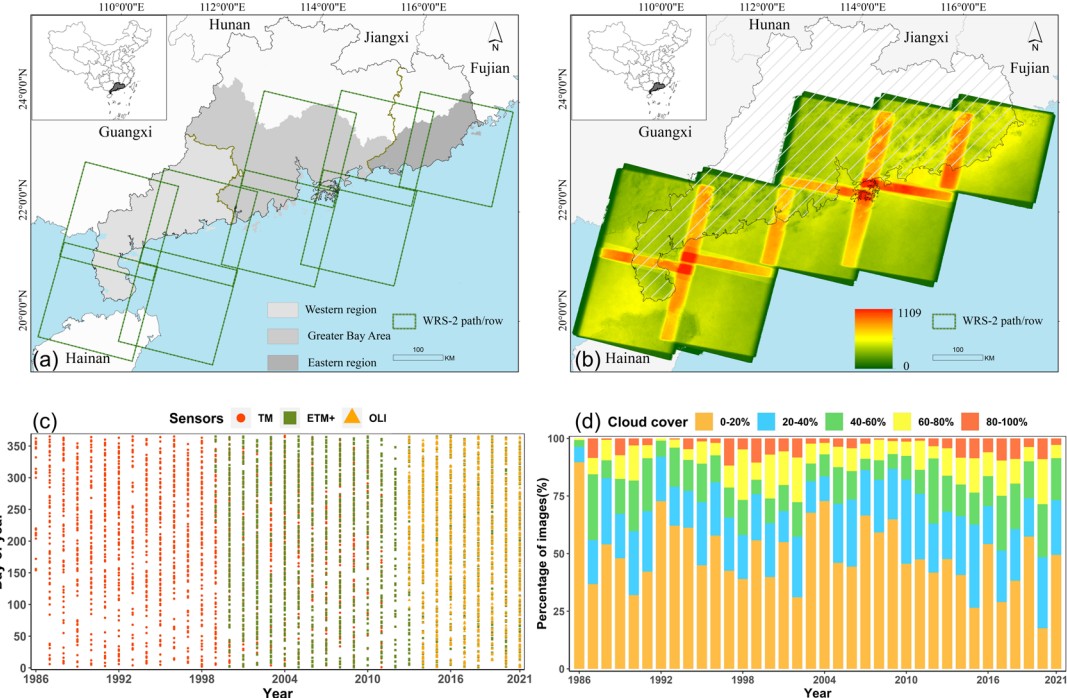

**Figure 1.** Location and the availability of Landsat images from 1986 to 2021 in the study area. (**a**) The geographical location of Guangdong, Hong Kong, and Macao, and nine tiles of Landsat Worldwide Reference System 2 (WRS-2) path/row coverage; (**b**) the spatial distribution of good-quality observation numbers at the pixel level; (**c**) the temporal distribution of Landsat images by sensors (TM, ETM+, and OLI) from 1986 to 2021; and (**d**) the proportion of Landsat images with different cloud covers from 1986 to 2021.

## 3. Methodology

We proposed an automated method for mapping the annual tidal flat changes in the micro-tidal areas based on the full-time series characteristics of dual spectral indices (Figure 2) and applied it in GHKM from 1986 to 2021. First, we implemented spectral harmonization for Landsat archive as data preprocessing (Section 3.1). Second, we selected the MNDWI and the NDWI as the dual spectral indices for tidal flat mapping by comparing the capabilities of multiple remote sensing indices (Section 3.2), combined the turning points from the dual spectral indices by a time series segmentation algorithm (Section 3.3), and developed a tide-independent hierarchical classification strategy based on inundation frequency and the Otsu algorithm (Section 3.4). Finally, we assessed the accuracy of the mapping results by visually interpreting samples in 1986, 2021, and the changed area, overlaying them with low-tide images, and comparing them with other available datasets (Section 3.5).

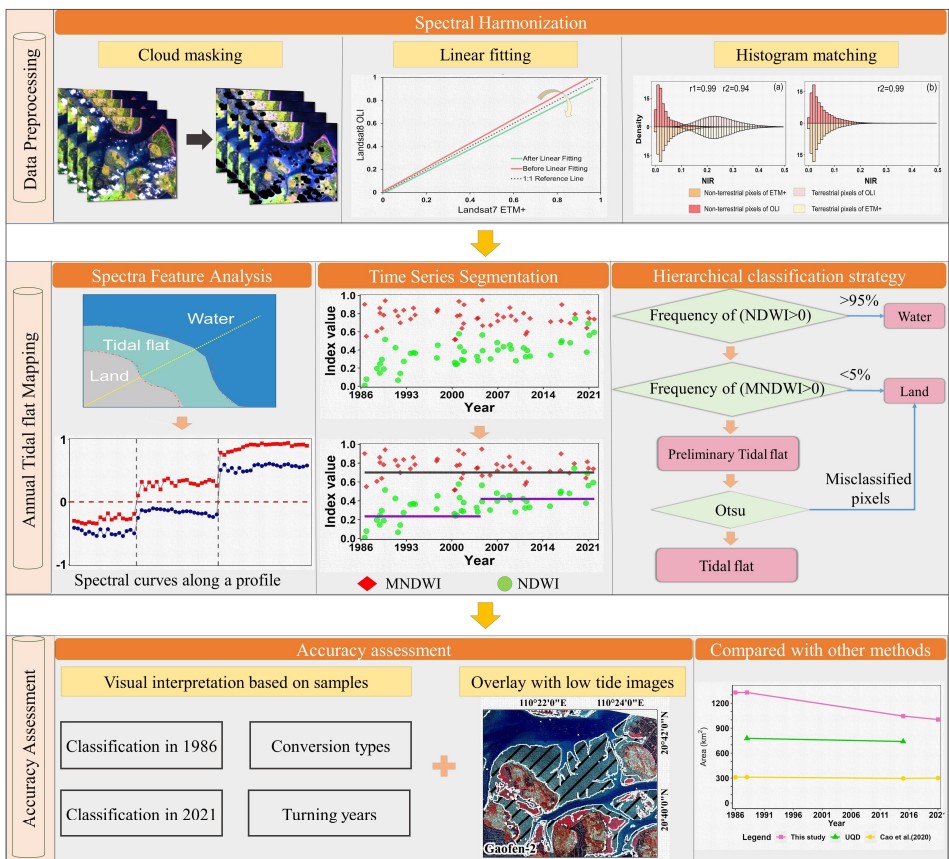

**Figure 2.** Framework for mapping long-term annual loss and gain in tidal flats.

### 3.1. Spectral Harmonization

The differences in surface reflectance spectra among different sensors should be harmonized prior to the long-term time series analyses, especially for the near-infrared (NIR) band [45]. We masked all the pixels flagged as cloud, cloud shadows by quality assessment (QA) band in image properties [46]. There are differences in spectra at bands of OLI and TM/ETM+ after cloud masking. Regarding this issue, we first applied the linear fitting equations proposed by Roy et al. [45] to adjust all the spectra bands of OLI sensors. These equations were established by fitting the ETM+ and corresponding OLI band values in the overlapped terrestrial vegetation area.

Roy's equations were developed in terrestrial areas, while this study focused on tidal flats that are periodically inundated by seawater. Therefore, after the first linear adjustment, we further check the consistency of corresponding bands between the OLI and ETM+ for both terrestrial and non-terrestrial pixels. We found that all the corresponding bands of the two sensors have a good agreement of spectral consistency, except the NIR band. Specifically, we calculated the Pearson correlation coefficients between the OLI and ETM+ corresponding bands by density values of each histogram bandwidth. The NIR band of the two sensors only have a good agreement for terrestrial pixels (r1 = 0.99), but not good enough for non-terrestrial pixels (r2 = 0.94) (Figure 3a), where r1 represents the correlation between ETM+ and OLI bands for terrestrial pixels; and r2 for non-terrestrial pixels. To address this issue, we further adjusted the NIR band spectra values of OLI for non-terrestrial pixels using the histogram matching method. Specifically, we first calculated the respective cumulative probabilities of the NIR band of OLI and ETM+, respectively, for all non-terrestrial pixels after 2013. Then, the NIR band values of OLI were adjusted using piecewise linear fitting based on the spectra values of ETM+ and the cumulative probabilities. The improvement in the correlation coefficient between the two sensors in the NIR band suggests the necessity and usefulness of the histogram matching method

(r2 = 0.99) (Figure 3b). All the harmonized Landsat time series data were used for the subsequent annual tidal flat mapping in this study.

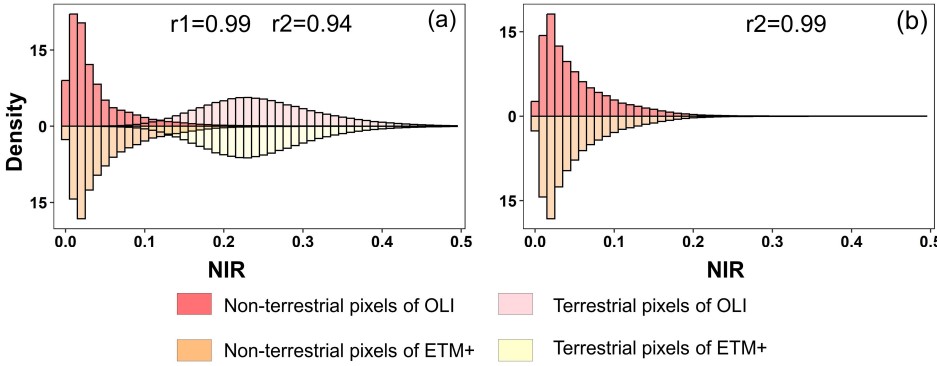

**Figure 3.** Histogram of NIR band of OLI and ETM+ sensors for non-terrestrial and terrestrial pixels. The r1 represents the correlation between ETM+ and OLI for terrestrial pixels, and r2 for non-terrestrial pixels. (**a**) The histogram of NIR for non-terrestrial and terrestrial pixels after the first linear adjustment; and (**b**) the histogram of NIR for non-terrestrial pixels after histogram matching.

*3.2. Spectra Feature Analysis*

We evaluated and compared the performances of four commonly used remote sensing spectral indices in distinguishing tidal flat, land, and water pixels. Specifically, tidal flat in this study refers to the non-vegetated tidal flat areas, land refers to the inland areas and the vegetated tidal flat areas (e.g., mangroves), and water refers to the offshore water areas. These spectral indices include the Normalized Difference Water Index (NDWI) calculated using Equation (1) [47], the Modified Normalized Difference Water Index (MNDWI) calculated using Equation (2) [48], the Land Surface Water Index (LSWI) calculated using Equation (3) [49], and the Automatic Water Extraction Index for shadow scenes (AWEIsh) calculated using Equation (4) [50], as they have been used in previous tidal flat mapping studies [51,52].

$$NDWI = \frac{GREEN - NIR}{GREEN + NIR} \tag{1}$$

$$MNDWI = \frac{GREEN - SWIR1}{GREEN + SWIR1} \tag{2}$$

$$LSWI = \frac{NIR - SWIR1}{NIR + SWIR1} \tag{3}$$

$$AWEIsh = BLUE + 0.25 \times GREEN - 1.5 \times (NIR + SWIR1) - 0.25 \times SWIR2 \tag{4}$$

where GREEN, NIR, SWIR1, SWIR2, and BLUE are surface reflectance in the green, near-infrared, shortwave infrared, and blue bands of the Landsat data, respectively.

We analyzed the spectral features of the four widely used spectral indices along profiles at 30 m intervals in three different representative locations abundant in tidal flat resources, i.e., Boti Harbor in the Western region (Figure 4a), Shenzhen Bay in the GBA (Figure 4b), and Zhelin Bay in the Eastern region (Figure 4c). This analysis suggested that the combination of the MNDWI and the NDWI with a robust threshold could well classify land, tidal flat, and water pixels in the micro-tidal area (Figure 4d). Specifically, for distinguishing the land and tidal flat pixels, the MNDWI performs well with a robust threshold of 0, while the AWEI and the NDWI could not distinguish using a stable threshold. The LSWI could not distinguish the tidal flat and land pixels due to their highly similar spectra values in the three profile locations. For distinguishing between tidal flat and water pixels at the seaward boundary, the NDWI performs well with a robust threshold of 0, while the AWEI, the LSWI, and the MNDWI could not distinguish effectively using a stable threshold, particularly in Shenzhen Bay and Zhelin Bay. In summary, the combination of

the NDWI and the MNDWI demonstrates significant spectral distinctions among land, tidal flat, and water pixels, contributing to distinguishing the boundaries of tidal flats (Figure 4d).

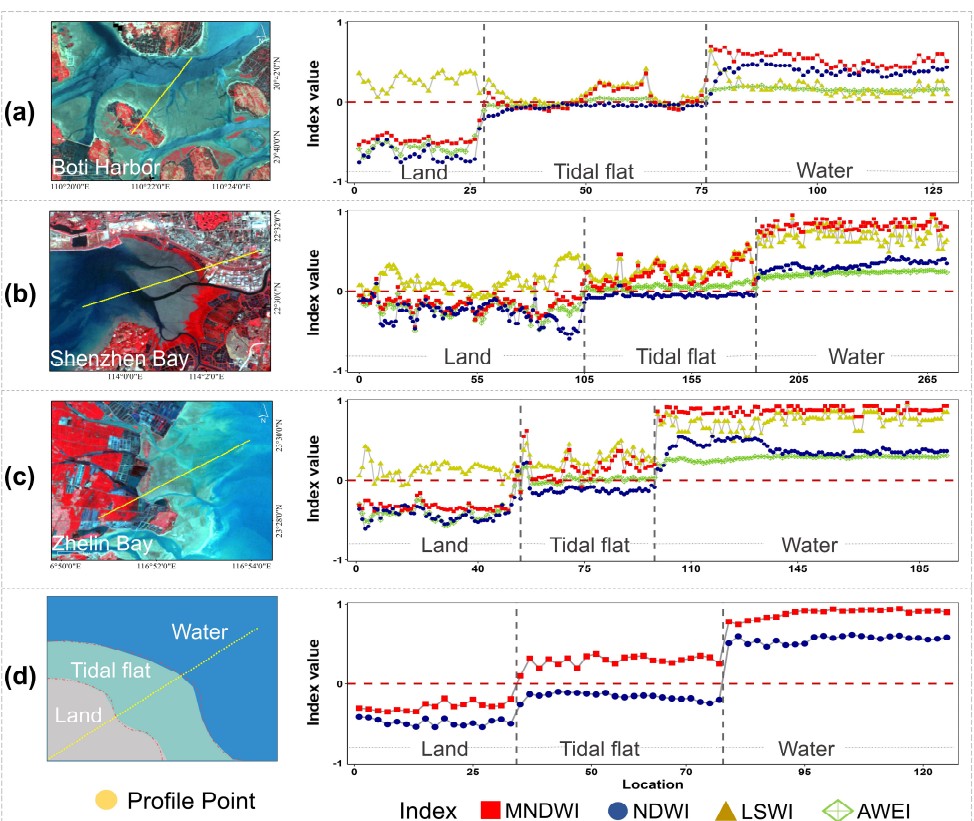

**Figure 4.** Spectra features of different indices along the profiles in (**a**) the Western region, (**b**) the GBA, and (**c**) the Eastern region, respectively; (**d**) a schematic diagram illustrates that the combination of the MNDWI and the NDWI can accurately map the boundaries of tidal flats with a robust threshold of 0.

### 3.3. Time Series Segmentation

We proposed the time series segmentation algorithm by detecting turning points with mean clustering in the full-time series of the NDWI and the MNDWI [53,54], to obtain the corresponding time series segments. The coastal cover types were assumed to be consistent within the same time series segment. Specifically, the one segment means that the pixel has the same coastal cover type in the entire time series. The two segments indicate that the pixel may have one change in coastal cover types. Multiple segments indicate that the pixel may have multiple changes in coastal cover types. We combined these turning points detected from the time series of the MNDWI and the NDWI to obtain all the time series segments and then merged the short time series segments with fewer than 10 observations or a time interval of less than 180 days into the adjacent and similar segments.

Two representative pixels in Pearl River Estuary were selected to illustrate the advantages of combining the MNDWI and the NDWI over using one index alone to capture tidal flat changes (Figures 5 and 6). The first pixel highlighted the greater capacity of the dual indices in capturing the tidal flat erosion compared to the MNDWI alone (Figure 5). The MNDWI failed to identify any turning points for this pixel, while the NDWI detected a turning point in the year of 2004 (Figure 5a). Based on the strategy of combining turning points, we obtained the turning point of 2004, as shown in the dash line (Figure 5a). The high- and low-tide images indicates that this pixel was inundated during high tide and exposed during low tide before 2004 (Figure 5b), while always inundated by water during

high and low tides after 2004 (Figure 5c), suggesting that this pixel converted from tidal flat to water in 2004.

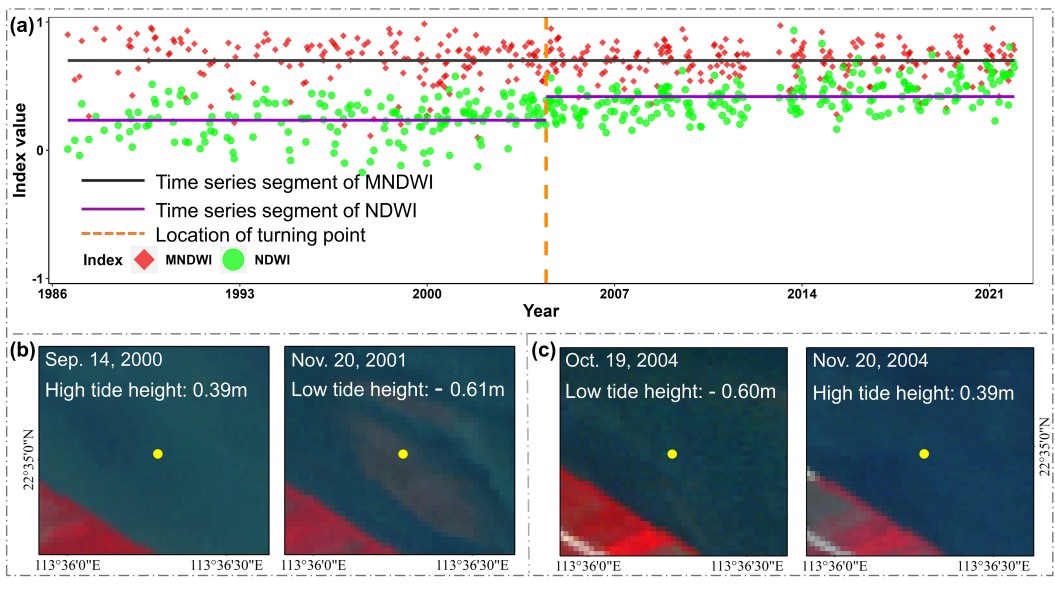

Figure 5. Time series segmentation for a pixel with tidal flat erosion in the year 2004 using the combination of the MNDWI and the NDWI. (**a**) The time series segments with a conversion from tidal flat to water in the turning year of 2004, (**b**) the two images show the tidal flat pixel inundated during high tide and exposed during low tide, corresponding to the left segments in (**a**), and (**c**) the two images show the water pixel inundated during both high and low tides, corresponding to the right segments in (**a**).

In addition, the pixel with multiple changes due to reclamation demonstrated a better capacity to capture changes using dual spectral indices rather than relying on one index alone (Figure 6). This pixel situated within the Tangjiawan reclamation project area of Zhuhai City, which underwent the significant land reclamation between 2000 and 2010. Specifically, the NDWI detected three turning years in 1999, 2008, and 2014, whereas the MNDWI detected those in 1998 and 2010, resulting in six initial time series segments (Figure 6a). After implementing the classification method detailed in Section 3.4, the adjacent time series segments with the same cover type were merged, resulting in the identification of two turning years in 1999 and 2010 (Figure 6b). The turning point of 1999 with the conversion from water to tidal flat was detected from the NDWI, and that of 2010 with the conversion from tidal flat to land was detected from the MNDWI. The Landsat images in the three rows illustrate the pixel conversions from water to tidal flat and subsequently to land due to land reclamation (Figure 6c), which is consistent with the land reclamation progress of Zhuhai City. In summary, the combination of the MNDWI and the NDWI proved to be effective in capturing more accurate changes in tidal flats.

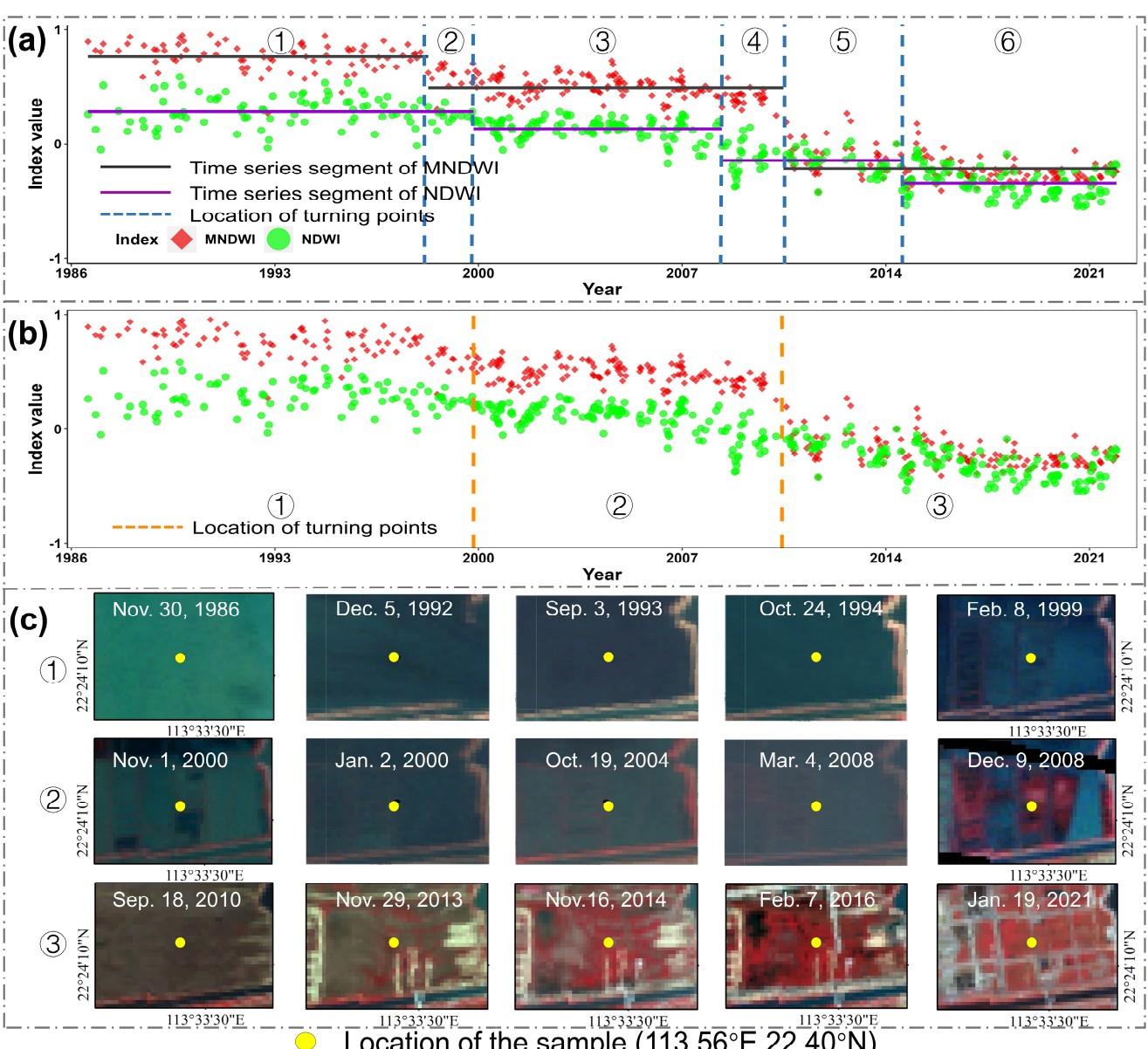

**Figure 6.** Time series segmentation for a pixel under reclamation using the combination of the MNDWI and the NDWI. (**a**) The initial six time series segments obtained and numbered ①–⑥; (**b**) the three time series segments numbered ①–③ with conversions from water to tidal flat and from tidal flat to land in the turning year of 1999 and 2010, respectively, after merging the adjacent segments with the same coastal cover types; and (**c**) the Landsat images in the ①–③ rows represent the water pixel, the tidal flat pixel, and the land pixel, corresponding to the ①–③ segments in (**b**).

### 3.4. Tide-Independent Hierarchical Classification Strategy

We proposed a tide-independent hierarchical classification strategy to identify conversion types. We first preliminarily classified the tidal flats based on the inundation frequencies of the dual-spectral indices, and then further removed misclassified tidal flats using the Otsu algorithm to extract the final tidal flat boundaries (Figure 7). This strategy does not rely on tide height data, making it more widely applicable for areas with microtidal ranges. Specifically, we first, respectively, calculated the inundation frequencies of the MNDWI and the NDWI in each time series segment. Based on the spectra feature analysis in Section 3.2, the spectral indices greater than 0 suggested that the pixels are inundated. Thus, we classified the pixels with inundation frequency of the NDWI greater than 95%

as water, the inundation frequency of the MNDWI below 5% as land, and the remaining pixels as preliminary tidal flat.

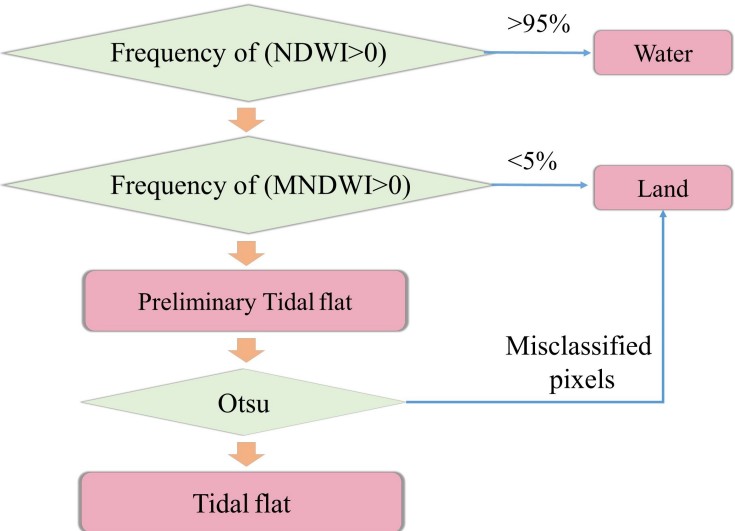

**Figure 7.** The tide-independent hierarchical classification strategy based on the inundation frequency and the Otsu algorithm.

However, certain coastal construction land and vegetation situated on tidal flats may be erroneously included in the preliminary tidal flat classification, by relying solely on inundation frequency for classification [8]. To address this, we utilized the Otsu algorithm to further remove these misclassified pixels. The Otsu is a threshold segmentation algorithm that could separate features of interest from the background by maximizing inter-class variance [55]. We obtained a threshold of 58% using the Otsu algorithm based on the inundation frequency of the MNDWI in the preliminary tidal flat area. The threshold of 58% is associated with the tidal flat features in GHKM. That is, tidal flats are relatively flat in GHKM because GHKM has a relatively micro-tidal height range and relatively small sediments discharged from rivers [31]. Further, the medium and high tidal flat areas are often covered by mangroves and salt marshes [35,56], so the non-vegetated tidal flat areas are mostly located in the low flat areas with high inundation frequency. Therefore, the reclassification threshold of 58% automatically obtained by the Otsu algorithm is reasonable in GHKM. We removed misclassified pixels with inundation frequency lower than 58% from the preliminary tidal flat area, obtaining the final tidal flat area.

After the classification procedures, we further removed the pseudo-turning points when the adjacent time series segments have the same coastal cover type. The removing was reasonable because there are no conversions of coastal cover types between the two adjacent time series segments. For instance, for the example pixel in Figure 6, we initially obtained six segments by combining the turning points detected from the NDWI and the MNDWI. By implementing the classification strategy, the coastal cover types of the first and second segments were water, the third and fourth were tidal flat, and the fifth and sixth were land. Thereby the pseudo-turning years in 1998, 2008, and 2014 were removed, and the real turning years in 1999 and 2010 were identified, as shown by the yellow dotted lines in Figure 6b. By iteratively removing all the pseudo-turning points, we obtained all the conversion types, turning years, and annual tidal flats at the pixel level. After obtaining the annual mapping results, we further automatically filled the morphologically enclosed patches in inland areas such as artificial lakes and aquaculture ponds.

### 3.5. Accuracy Assessment Strategy

Annual accuracy of tidal flats was assessed through three aspects, including visually interpreting the validation samples to establish ground truth reference, overlaying the

results with low-tide images, and comparing this study with other methods. Specifically, we randomly sampled 200 pixels from each land, tidal flat, and water area in both 1986 and 2021, respectively, to assess the accuracy of the initial and final coastal cover type classification. Meanwhile, we randomly sampled 400 pixels in the changed area, 72% of which represented single changes and 28% represented multiple changes, thus, a total of 545 conversions from the 400 pixels were used to assess the accuracy of conversion types and turning years (Figure 8). Further, we qualitatively assess the tidal flat boundaries by overlaying them with low tide satellite images and comparing them with the other methods [8,26].

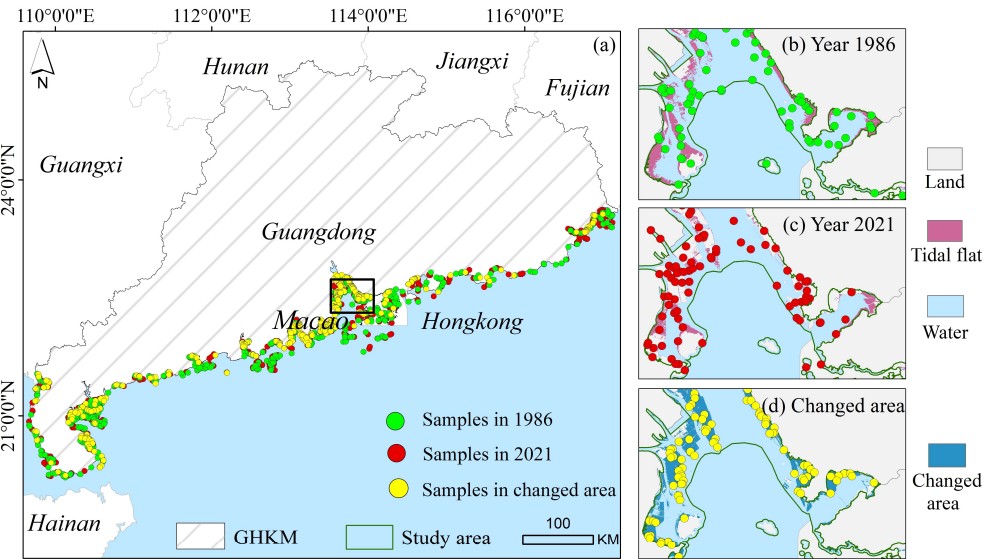

**Figure 8.** (**a**) Spatial distribution of validation samples in the study area. Zoom-in maps of (**b**) samples in 1986, (**c**) samples in 2021, and (**d**) samples in the changed area in the Pearl River Estuary.

## 4. Results

### 4.1. Accuracy of Annual Tidal Flats

The accuracy assessment results suggest that the proposed method performed well in identifying tidal flats loss and gain at the annual interval (Table 1, Figures 9 and 10). The method achieved an overall accuracy (OA) of 95% and a kappa coefficient of 0.92 in coastal cover type classification in 1986 (Table 1a), as well as an OA of 93% and a kappa coefficient of 0.90 in 2021 (Table 1b). Meanwhile, in 1986, the user's accuracy (UA) and producer's accuracy (PA) were from 89% to 99% and 90% to 98%, respectively. In 2021, the UA were from 86% to 97%, and the PA were from 89% to 97%. The classification errors in both 1986 and 2021 were primarily from the confusions between tidal flat and water pixels, due to the high suspended sediment content in the Pearl River Estuary and pixel spectral mixing issues along the rocky coast. The high suspended sediment content in the Pearl River Estuary, coupled with the spectral similarity of the seaward boundary of micro-tidal zones' tidal flats, has led to partial misclassification of suspended sediment in our method while capturing more comprehensive tidal flat boundaries. Despite of this error, our classification accuracy remains relatively high.

**Table 1.** Confusion matrix tables of coastal cover types in (a) 1986, (b) 2021, and (c) conversion types.

| a (1986) | | Reference | | | |
|---|---|---|---|---|---|
| | | Land | Tidal flat | Water | UA |
| | Land | 194 | 4 | 2 | 97% |
| Classification | Tidal flat | 4 | 177 | 19 | 89% |
| | Water | 0 | 2 | 198 | 99% |
| | PA | 98% | 97% | 90% | |
| | OA | 95% | Kappa | 0.92 | |

| b (2021) | | Reference | | | |
|---|---|---|---|---|---|
| | | Land | Tidal flat | Water | UA |
| | Land | 194 | 6 | 0 | 97% |
| Classification | Tidal flat | 6 | 171 | 23 | 86% |
| | Water | 1 | 6 | 193 | 97% |
| | PA | 97% | 93% | 89% | |
| | OA | 93% | Kappa | 0.90 | |

| c (Conversion types) | | | | Reference | | | |
|---|---|---|---|---|---|---|---|
| Classification | Land to Tidal flat | Land to Water | Tidal flat to Land | Tidal flat to Water | Water to Land | Water to Tidal flat | Unchanged |
| Land to Tidal flat | 29 | 5 | 0 | 0 | 0 | 0 | 4 |
| Land to Water | 1 | 19 | 0 | 3 | 0 | 0 | 0 |
| Tidal flat to Land | 0 | 0 | 187 | 0 | 6 | 0 | 10 |
| Tidal flat to Water | 0 | 1 | 0 | 67 | 0 | 0 | 15 |
| Water to Land | 0 | 0 | 0 | 0 | 104 | 1 | 0 |
| Water to Tidal flat | 0 | 0 | 0 | 0 | 1 | 78 | 11 |
| Unchanged | 0 | 0 | 0 | 0 | 0 | 0 | 11 |
| | OA | 89% | | Kappa | 0.86 | | |

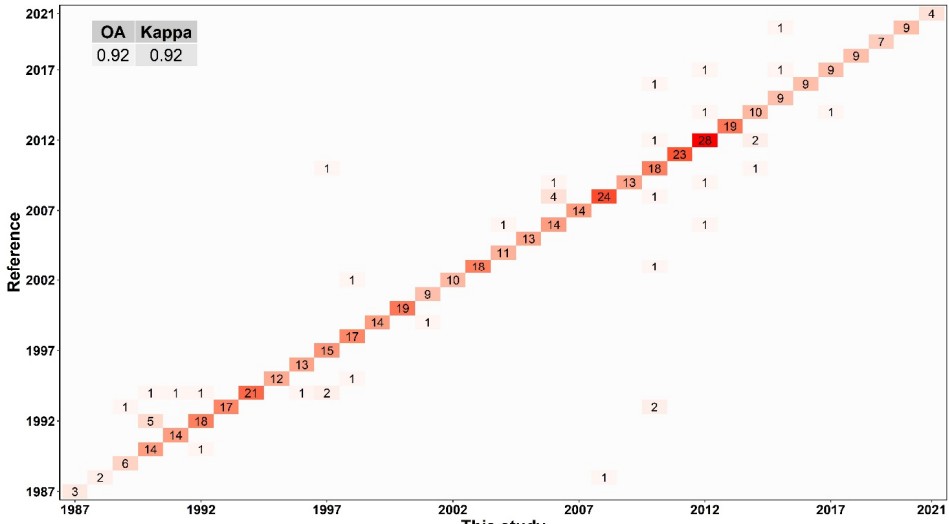

**Figure 9.** Accuracy assessment of turning years during 1987–2021. The numbers represent the sample numbers in different years. Darker color indicates larger sample number.

The method also shows good performance in capturing inter-annual tidal flat changes, with the OA of 89% for conversion types and 92% for turning years (Table 1c and Figure 9). The majority of the samples are located on the diagonal with one-year tolerance, suggesting that the turning years in this study are consistent with the references from visual interpretation. Our method utilizes all available time series images at the pixel level, ensuring each pixel has adequate observations to map annual continuous changes and minimize the impact of cloud cover and atmospheric conditions on the accuracy. The errors of conversion

types primarily came from the aforementioned classification confusion between tidal flat and water pixels.

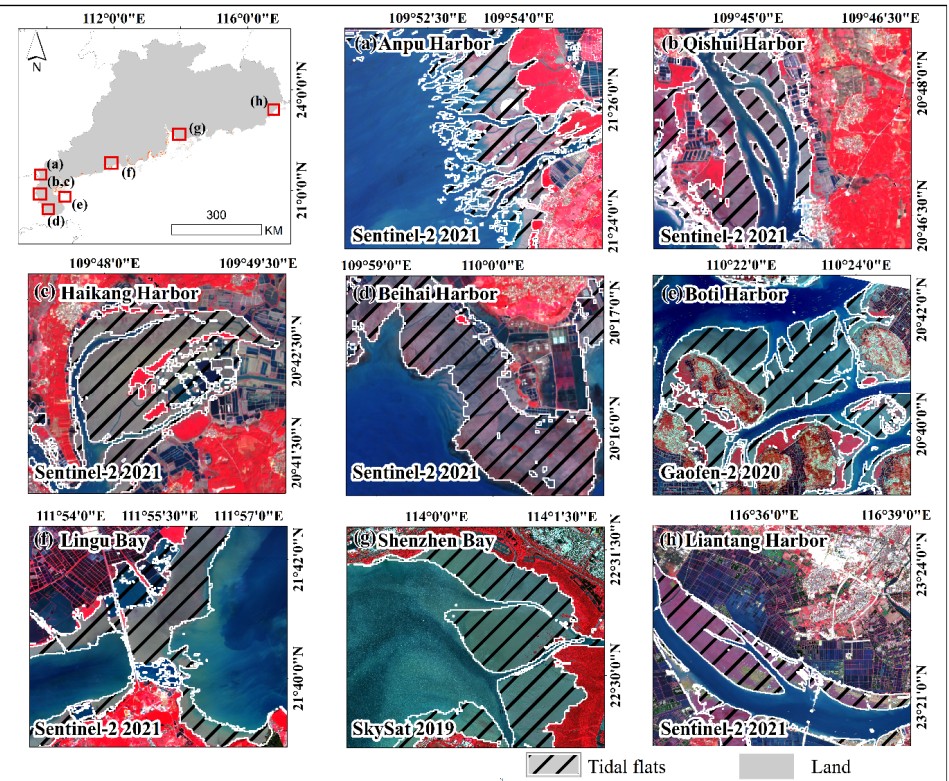

**Figure 10.** The mapped tidal flats in GHKM overlay with Gaofen-2, SkySat, and Sentinel-2 low-tide images. There are eight sites: (**a**) Anpu Harbor, (**b**) Qishui Harbor, (**c**) Haikang Harbor, (**d**) Beihai Harbor, (**e**) Boti Harbor, (**f**) Lingu Bay, (**g**) Shenzhen Bay, and (**h**) Liantang Harbor.

The mapped tidal flats in eight local coastal regions abundant in tidal flat resources were overlaid on low tide satellite images including Sentinel-2 (spatial resolution of 10 m), GF-2 (spatial resolution of 3.2 m), and SkySat (spatial resolution of 0.5 m) (Figure 10). The seaward and landward boundaries of tidal flats in these regions were consistent with the low tide satellite images. Specifically, the seaward boundaries of the tidal flats in these areas are well consistent with the area exposed on the satellite images. The landward boundaries between tidal flats and mangroves were well distinguished in Anpu Harbor (Figure 10a), Qishui Harbor (Figure 10b), Haikang Harbor (Figure 10c), Boti Harbor (Figure 10e), and Shenzhen Bay (Figure 10g). Further, the landward boundaries between tidal flats and aquaculture pond patches were also well distinguished in Beihai Harbor (Figure 10d), Lingu Bay (Figure 10f), and Liantang Harbor (Figure 10h).

### 4.2. Annual Loss and Gain Dynamics of Tidal Flats

The annual loss and gain of tidal flats from 1986 to 2021 at 30 m resolution in GHKM were obtained using the proposed automated method. The results show that the tidal flats in GHKM were 1328.74 km$^2$ in 1986 and 1004.17 km$^2$ in 2021, with a decrease of 24% (Figure 11a). More than 70% of the tidal flat loss was attributed to direct human activities such as aquaculture expansion and land reclamation. Of the total tidal flat loss, 504.45 km$^2$ have been lost due to land reclamation and aquaculture pond expansion and only 179.88 km$^2$ have been gained along the newly formed coastlines. In addition, the dominant conversion types are the conversions from tidal flat and water to land caused by land reclamation (Figure 11b). Specifically, the newly added land area reached 812.33 km$^2$ during the entire study period by encroaching tidal flat (66%) and water areas (34%) (Figure 11b).

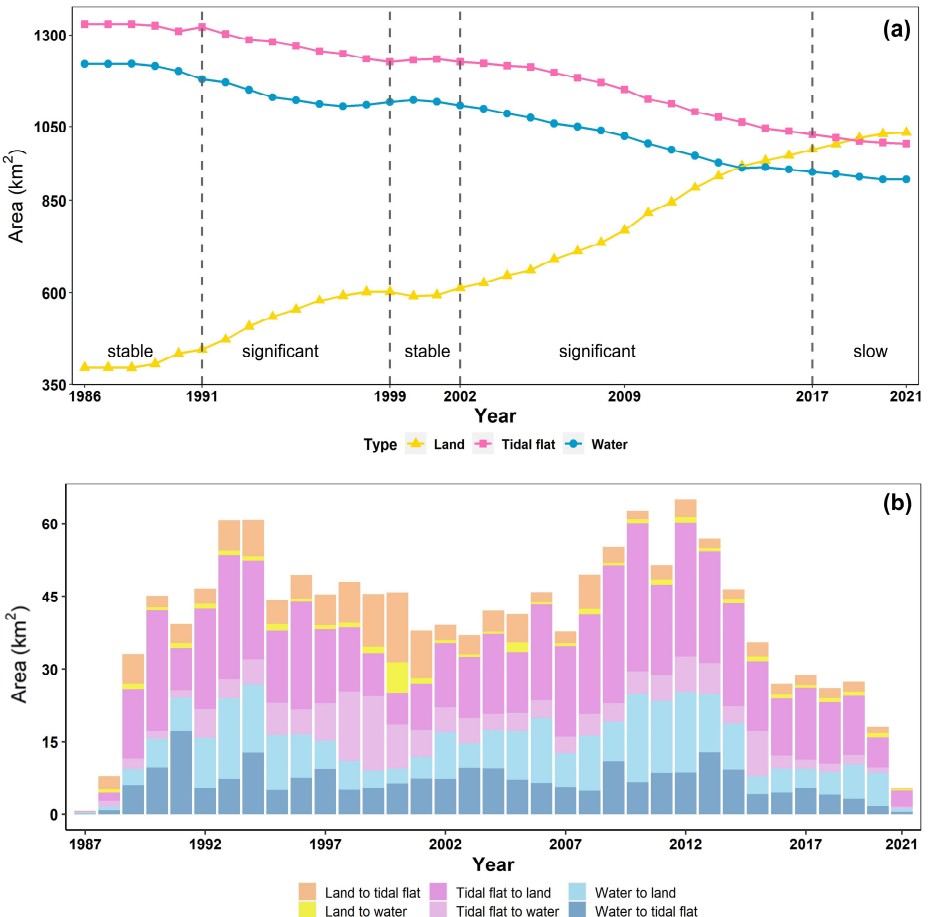

**Figure 11.** Annual statistic areas of coastal cover types and tidal flat changes in GHKM. (**a**) Annual area of three coastal cover types and the five divided periods; and (**b**) annual area of conversion types during 1986–2021.

There are five sub-periods according to the temporal loss rates of tidal flats in GHKM, i.e., two stable periods (1986–1990; 1999–2001) both with annual average change rates of 0.3%, two significant change periods (1991–1998; 2002–2016) with annual average change rates of 1.0% and 1.2%, and a slow change period (2017–2021) with an annual average change rate of 0.6% (Figure 11a). Specifically, during the two stable change periods, the tidal flats has relatively slight fluctuations due to the small amount of land reclamation activities. During the slow change period from 2017 to 2021, the tidal flats accreted by 22.76 km$^2$ but were encroached by 56.35 km$^2$, resulting in an annual loss of 8.40 km$^2$/y. The loss of tidal flats during the slow change period accounted for 7.6% of the total loss during the entire study period. During the first significant change period from 1991 to 1998, the tidal flats accreted by 115.69 km$^2$ and were encroached by 192.00 km$^2$, with an annual loss of 10.90 km$^2$/y (Figure 11a). Similarly, during the second significant change period from 2002 to 2016, the tidal flats accreted by 166.62 km$^2$ and were encroached by 362.36 km$^2$, with an annual loss of 14.01 km$^2$/y (Figure 11a). The loss of tidal flats during the two significant change periods accounted for 85% of the total loss during the entire study period (Figure 11a).

Regarding the spatial distribution, the tidal flat loss was concentrated in the GBA, followed by the Western region, while the Eastern region is relatively well conserved (Figure 12a–c). Meanwhile, due to the accretion of tidal flats along the new reclaimed coastline in the GBA and migration of some sandbars in the Western region, there were a few instances of tidal flat gain. Furthermore, we quantified the tidal flat area changes in the sub-regions, i.e., the Western region, the GBA and the Eastern region (Figure 12d–g). By 2021, 48.8% of the total tidal flat area were in the Western region, 43.0% in the GBA, and

8.2% in the Eastern region, respectively (Figure 12d). Specifically, throughout 1986–2021, the tidal flats decreased by 230.25 km² with a decrease of 35.5% in the GBA (Figure 12f), by 79.18 km² with a decrease of 13.9% in the Western region (Figure 12e), and by 15.15 km² with a decrease of 13.9% in the Eastern region (Figure 12g), respectively. Overall, 70.9% of the total loss was in the GBA, 24.4% in the Western region, and 4.7% in the Eastern region during the entire study period (Figure 12d). Consequently, the GBA plays a dominant role in tidal flat loss during the entire period.

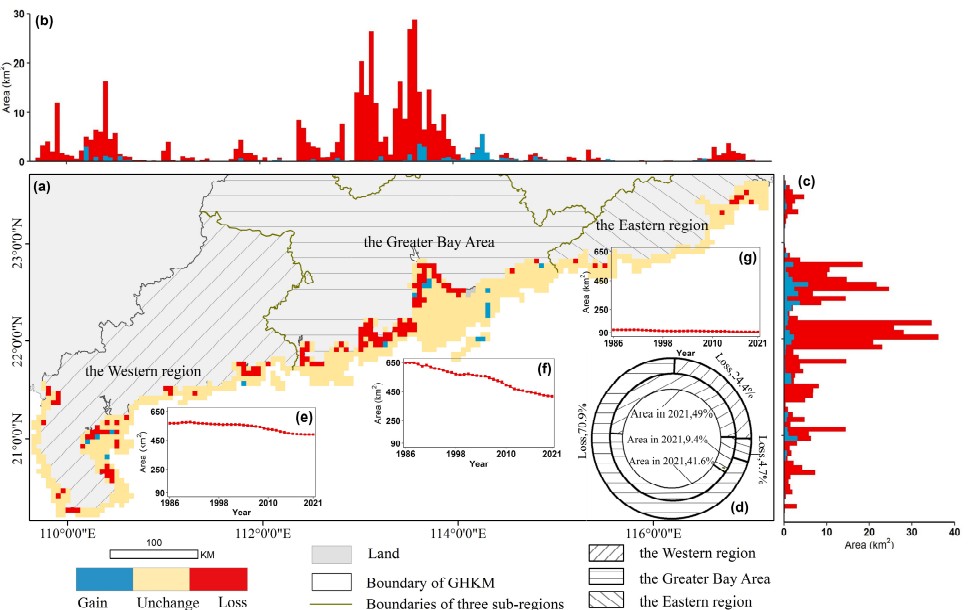

**Figure 12.** (**a**) The tidal flat loss and gain from 1986 to 2021 per 0.05° grid in GHKM; summaries of tidal flat loss and gain along the (**b**) longitude and (**c**) latitude coordinates; (**d**) the proportion of tidal flat area in 2021 and loss from 1986 to 2021 for three sub-regions; the annual area of tidal flats in (**e**) the Western region, (**f**) the Greater Bay Area, and (**g**) the Eastern region.

Here, we showed the typical areas with significant coastal changes in the GBA (Figure 13). Specifically, Zhuhai and Shenzhen, among the earliest special economic zones in China, experienced substantial land reclamation for port construction before 2009 (Figure 13a,b). The finding is consistent with the findings by Ai et al. [57] on land reclamation activities in Guangdong Province. In Macao, the tidal flats between Taipa and Coloane islands were filled to land for urban expansion from 1987 to 2005, and the major artificial island project for the Zhuhai–Macao Bridge, was reclaimed from water areas between 2009 and 2013 (Figure 13c). It only took about 20 years for the area of Macao to more than double (Figure 13c). In Hong Kong, we found that the Hong Kong Airport Core Program was finished in 1997 by encroaching water (Figure 13d). Additionally, to support the construction of the Hong Kong–Zhuhai–Macao Bridge, an artificial island was constructed by encroaching water areas in Chek Lap Kok between 2014 and 2016 (Figure 13d). This suggested that, land reclamation is mainly realized by encroaching water areas due to limited tidal flat areas in Hong Kong. The finding is consistent with the progress of coastal land reclamation projects in Hong Kong [58].

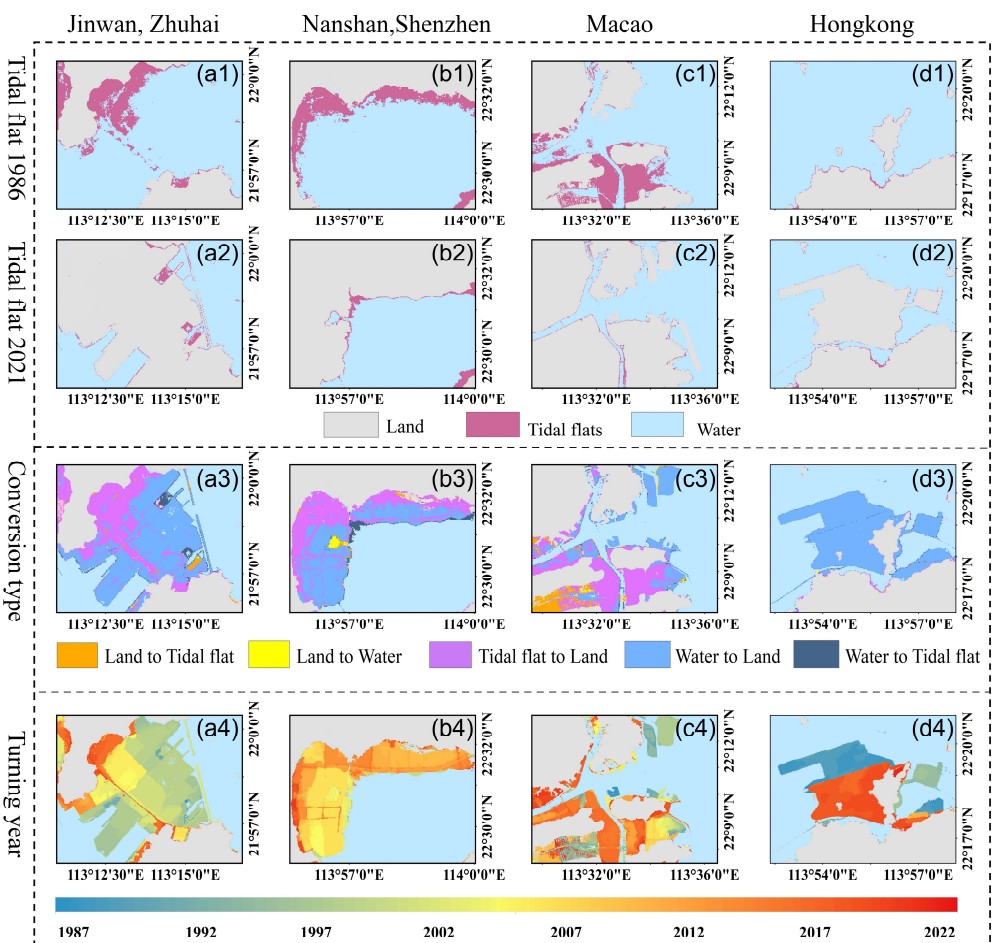

**Figure 13.** Four typical regions with significant coastal changes. The (**a1–d1**) are the tidal flat spatial distribution in 1986, the (**a2–d2**) are in 2021. The (**a3–d3**) present the conversion types, and the (**a4–d4**) present the turning years.

### 4.3. Policy Drivers for the Tidal Flat Dynamics

The periods of 1991–1998 and 2002–2016 were marked by significant tidal flat loss. The loss could be attributed to various policies implemented by the national, provincial, and local governments. After the reform and opening up, government have actively promoted the development of aquaculture in response to the increasing seafood demand. As a result, there has been an expansion of aquaculture areas under the guidance of Instruction on Relaxing Policies and Accelerating the Development of Aquaculture, leading to the encroachment on tidal flats [59,60]. During 1991–1998, the loss of tidal flats was in the special economic zones of Zhongshan and Zhuhai Cities in Guangdong, as well as the special administrative region Macao. These regions have carried out aquaculture and agriculture expansion activities by encroaching the tidal flats and waters, such as paddy fields, cropland, and fishponds, to support local population and industrial development [61,62]. For example, reclamations for developing agriculture were carried out around the Hengqin, Zhuhai City since it was designated as a key development zone in 1992. Overall, coastal changes in the 1990s were dominated by aquaculture expansion, which consistent with other China's provinces such as Zhejiang and Fujian [8,14].

In addition, the tidal flat loss during the period from 2002–2016 is more widespread compared to the 1990s. Specifically, in 2002, the enactment of the Sea Area Use Management Law of the People's Republic of China provided a legal basis for activities such as land reclamation in coastal areas. Subsequently, under the guidance of the National Marine Economic Development Plan in 2003 [63], Guangdong province carried out widespread land reclamation for the construction of industrial development zones, seaside ecological travel

areas, and urban infrastructure. Since then, there has been a steady annual increase in the area of tidal flat encroachment. As one of the national marine economic development pilot areas established in 2011, reclamation activities were concentrated in the GBA including Nansha District of Guangzhou, Jiaoyi Bay of Dongguan City, Hengmen of Zhongshan City, Shenzhen, and Hengqin of Zhuhai for ports and coastal industries. Such high-intensity reclamation projects continued until 2016.

The annual loss rates of tidal flats began to slow down since 2017. This shift may be attributed to the Notice of the State Council on Strengthening Coastal Wetland Protection and Strict Control of Reclamation promulgated in 2018 [62], which lead to the strict control of reclamations activities and the protection of coastal zone resources. Further, GHKM has been listed as the key area of the China's master plan (2021–2035) for major projects to protect and restore key ecosystems. It can be expected that after 2021, the tidal flat resources in the study area will undergo ecological restoration and increase in area. This study successfully reviewed the historical trajectories of tidal flats in GHKM over three decades and identified the significant loss regions, which could serve as basis for the site selection of coastal restoration projects.

## 5. Discussion

### 5.1. Mapping Performances Compared with Other Methods

We compared our mapping results with University of Queensland global tidal flats dataset (UQD) maps developed by Murray et al. [26], as well as the results using the method of Cao et al. [8]. We found that this study captured more accurate landward and seaward boundaries of tidal flats and more reliable tidal flat changes (Figure 14).

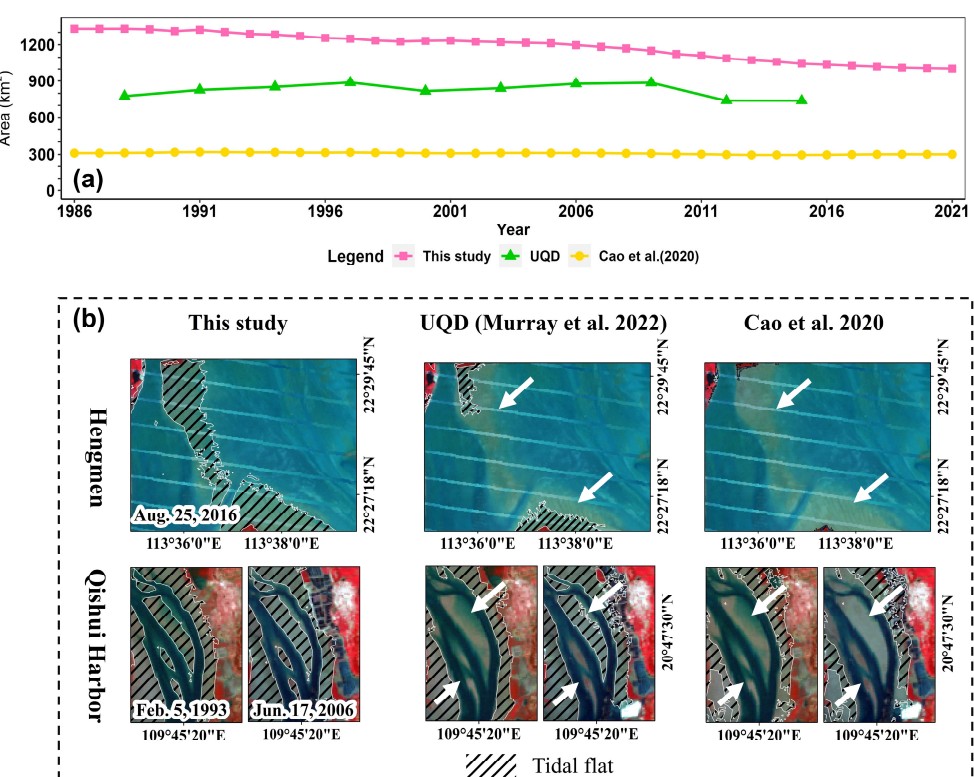

**Figure 14.** (**a**) Comparison of the statistical areas and mapping results of tidal flats among this study, Murray et al. [26], and Cao et al. [8]. The location of the typical areas in (**b**) can be found in Figure 10a.

Currently, the publicly available tidal flat dataset in GHKM is only the UQD maps [26]. This method uses the 3-year image stack and random forest for tidal flat classification. We found that there are underestimation and pseudo-interannual variations in the tidal flat area by UQD (Figure 14a). For example, we compared our mapping results with the UQD

map in the Hengmen in Pearl River Estuary and in Qishui Harbor in the Western through overlaying with the low-tide images in the corresponding years (1993, 2006, and 2016) (Figure 14b). In Hengmen, most of the tidal flats cannot be mapped in UQD, resulting in a considerable underestimation. In Qishui Harbor, the tidal flat area remained unchanged from 1993 to 2006, but the results from UQD in 1993 and 2006 were inconsistent, leading to pseudo-increase during this period. This has resulted in a bias in the annual statistical change area trend of UQD in GHKM (Figure 14a). Compared to UQD's method, our approach adopts flexible temporal segments for each pixel based on the full-time series spectral variation characteristics, instead of solely using a fixed period window, thus helping to avoid the underestimation and pseudo-interannual variations in tidal flats.

Further, we compared this study with the method also adopting the flexible temporal segments [8]. We found that the method of Cao et al. [8] obtained very limited annual tidal flat areas and even almost no tidal flat changes in GHKM (Figure 14a). For instance, areas such as Hengmen at the Pearl River Estuary and Qishui Harbor in the Western region showed significant underestimations in the mapped tidal flat seaward boundary (Figure 14b). This is due to the weak ability of the MNDWI to map the seaward boundary of micro-area tidal flats, which is consistent with our findings in Section 4.2.

Overall, the proposed method offers several advantages: it is independent of training samples, automated, and applicable for large-scale areas. Additionally, it enables the identification of continuous change trends in tidal flats without relying on cloud-free images or tide-level data. Importantly, our analysis demonstrates that our approach accurately delineates the landward and seaward boundaries of tidal flats and provides more reliable information regarding tidal flat changes by mitigating underestimation and pseudo-interannual variations.

### 5.2. Sensitivity Analysis of the Method

We explored the benefits of the further spectral harmonization of the NIR band by comparing the accuracy of conversion types and turning years (Tables 2 and 3, Figure 15a). We found spectral differences in the NIR band between OLI and ETM+ in the non-terrestrial areas after Roy et al. [45]'s linear equations and further performed spectral harmonization using histogram matching method before long-term annual tidal flat mapping in Section 3.1. Here, we obtained annual tidal flats using all the same methodology except the histogram matching. Without spectra harmonization by histogram matching led to the decreases by 3% and 1% of OA for conversion types and turning years, respectively (Tables 2 and 3, Figure 15a). The differences in turning years were concentrated after 2013 due to the differences in spectra at NIR band between OLI and TM/ETM+. Specifically, after 2013, four wrong conversions from tidal flat to water were added because of the higher NDWI values calculated by the non-harmonized NIR band values from OLI (Table 3, Figure 15a). The findings suggest that it is essential to reduce the spectral differences by further spectra harmonization of NIR among sensors for time series tidal flat mapping.

**Table 2.** The performance of different strategies.

|  | OA of Conversion Types | OA of Turning Years |
| --- | --- | --- |
| This study | 89% | 92% |
| Without histogram matching | 86% | 91% |
| Using the OSTU threshold of 63% | 88% | 91% |
| Using the OSTU threshold of 53% | 88% | 91% |

**Table 3.** Confusion matrix tables of conversion types after 2013 without histogram matching.

| Without Histogram Matching | Reference | | | | | | |
|---|---|---|---|---|---|---|---|
| Classification | Land to Tidal flat | Land to Water | Tidal flat to Land | Tidal flat to Water | Water to Land | Water to Tidal flat | Unchanged |
| Land to Tidal flat | 4 (−1) | 0 | 0 | 0 | 0 | 0 | 0 |
| Land to Water | 1 (+1) | 2 | 0 | 1 | 0 | 0 | 0 |
| Tidal flat to Land | 0 | 0 | 41 (−1) | 0 | 1 | 0 | 3 (+1) |
| Tidal flat to Water | 0 | 0 | 0 | 11 (+1) | 0 | 0 | 6 (+4) |
| Water to Land | 0 | 0 | 1 (+1) | 0 | 20 | 0 | 0 |
| Water to Tidal flat | 0 | 0 | 0 | 0 | 0 | 9 | 2 |
| Unchanged | 0 | 0 | 0 | 0 (−1) | 0 | 0 | 0 (−5) |
| | OA | 86% | | | | | |

Note: The numbers indicate count of samples. The numbers in red are the differences in sample counts corresponding to the decrease in accuracy, and the numbers in green are the differences in sample count for the improvement in accuracy, compared with this study.

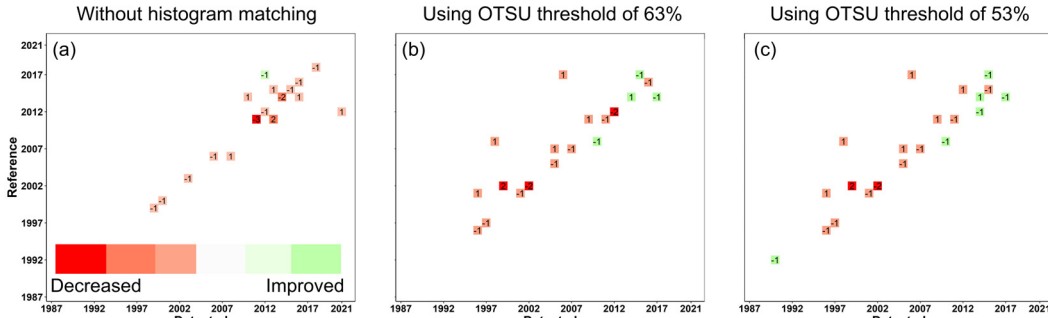

**Figure 15.** The differences in turning years between this study and those without histogram matching (**a**), and using thresholds of 58% ± 5% (**b**,**c**). Note: The numbers indicate the count of samples. The numbers in red are the differences in sample counts corresponding to the decrease in accuracy, and the numbers in green are the differences in sample count for the improvement in accuracy, compared with this study.

A sensitivity analysis was implemented using thresholds of 58% ± 5% to explore the influences of reclassification threshold in tidal flat mapping accuracy (Tables 2 and 4, Figure 15). The inundation frequency threshold of the MNDWI (i.e., 58%) based on the Otsu algorithm was used for tidal flat reclassification in Section 3.4. When using the threshold of 63% and 53%, there was little difference in the accuracies with only decreases by 1% in OA for both conversion types and turning years (Tables 2 and 4, Figure 15). Specifically, when using the Otsu threshold of 63%, a very small number of tidal flats might be misclassified as land (Table 4a), and when using 53%, a very small number of land pixels might be misclassified as tidal flats (Table 4b). Overall, the Otsu threshold value had a limited impact on the accuracy of tidal flat mapping. The above sensitivity analyses suggest that our method shows great robustness and could be extended to other large-scale regions.

**Table 4.** Confusion matrix tables of conversion types using Otsu thresholds of 63% (a) and 53% (b), respectively.

| (a) Using Otsu threshold of 63% | | | | Reference | | | |
|---|---|---|---|---|---|---|---|
| Classification | Land to Tidal flat | Land to Water | Tidal flat to Land | Tidal flat to Water | Water to Land | Water to Tidal flat | Unchanged |
| Land to Tidal flat | 29 | 5 | 0 | 0 | 0 | 0 | 41 |
| Land to Water | 1 | 19 | 0 | 4 (+1) | 0 | 0 | 0 |
| Tidal flat to Land | 0 | 0 | 183 (−4) | 0 | 6 | 0 | 10 |
| Tidal flat to Water | 0 | 1 | 0 | 66 (−1) | 0 | 0 | 15 |
| Water to Land | 0 | 0 | 3 | 0 | 104 | 4 (+3) | 0 |
| Water to Tidal flat | 0 | 0 | 0 | 0 | 1 | 75 (−3) | 11 |
| Unchanged | 0 | 0 | 4 (+4) | 0 | 0 | 0 | 11 |
| | OA | 88% | | | | | |
| (b) Using Otsu threshold of 53% | | | | Reference | | | |
| Classification | Land to Tidal flat | Land to Water | Tidal flat to Land | Tidal flat to Water | Water to Land | Water to Tidal flat | Unchanged |
| Land to Tidal flat | 29 | 5 | 0 | 0 | 0 | 0 | 4 |
| Land to Water | 1 | 19 | 0 | 4 (+1) | 0 | 0 | 0 |
| Tidal flat to Land | 0 | 0 | 183 (−4) | 0 | 6 | 0 | 10 |
| Tidal flat to Water | 0 | 1 | 0 | 66 (−1) | 0 | 0 | 15 |
| Water to Land | 0 | 0 | 3 | 0 | 104 | 1 | 0 |
| Water to Tidal flat | 0 | 0 | 0 | 0 | 1 | 78 | 11 |
| Unchanged | 0 | 0 | 4 (+4) | 0 | 0 | 0 | 11 |
| | OA | 88% | | | | | |

Note: The numbers indicate count of samples. The numbers in red are the differences in sample counts corresponding to the decrease in accuracy.

## 6. Conclusions

In this study, we mapped the long-term annual tidal flat changes in GHKM integrating dual time series spectral indices from Landsat during 1986–2021 without the dependence on tidal height data. The proposed method is independent of training samples, automated, and applicable for large-scale areas. Specifically, we harmonized the spectral consistency for different Landsat sensors, proposed a combination of dual spectral indices by profile-based spectral analysis in the micro-tidal area, and performed the time series segmentation. Furthermore, we identified the annual tidal flat loss and gain dynamics by developing a tide-independent hierarchical classification strategy based on inundation frequency and the Otsu algorithm.

Our approach revealed the annual dynamics of tidal flats in the GHKM from 1986–2021. The OA for conversion types and turning years achieved 89% and 92%, respectively. The tidal flats in GHKM decreased from 1328.74 km$^2$ in 1986 to 1004.17 km$^2$ in 2021. Of the total decrease, 504.45 km$^2$ have been lost due to land reclamation and aquaculture pond expansion and only 179.88 km$^2$ have been gained along the newly formed coastlines. There are five sub-periods according to the change rates of tidal flats in GHKM, i.e., two stable periods (1986–1990; 1999–2001) both with annual average change rates of 0.3%, two significant change periods (1991–1998; 2002–2016) with annual average change rates larger than 1.0%, and a slow change period (2017–2021) with an annual average change rate of 0.6%. The tidal flat loss was concentrated in the GBA, followed by the Western region, while the Eastern region is relatively well conserved.

We found that this study captured more accurate landward and seaward boundaries of tidal flats and more reliable tidal flat changes compared with UQD maps [26] and Cao et al. [8]'s method. Additionally, we found that spectral harmonization for the NIR band spectra is beneficial to tidal flat mapping with an increase by 3% in OA for conversion

types and a 1% increase for turning years. The threshold value from Otsu had a limited impact on the accuracy of tidal flat changes.

This study offers several avenues for future studies. First, the findings from this study could serve as basis for the site selection of coastal restoration projects, by identifying the hotpot areas of historical tidal flat loss. Second, the accurate boundaries of tidal flats could serve as the foundation for various in-depth studies including tidal flat elevation inversion, assessment of ecosystem services, and estimation of carbon storage. Third, it suggests that the proposed method is promising to be extended to other large-scale areas. Investigating the method's applicability across diverse global regions presents a promising direction for future research.

**Author Contributions:** Conceptualization, H.Z., W.C. and J.L.; methodology, H.Z., W.C. and J.L.; writing—original draft preparation, J.L., W.C. and H.Z.; writing—review and editing, J.L., W.C. and H.Z.; review, X.L., Y.Z., S.H., Z.Z. and D.L. All authors have read and agreed to the published version of the manuscript.

**Funding:** This research was funded by the National Natural Science Foundation of China (No. 42106174) and the open fund of the State Key Laboratory of Satellite Ocean Environment Dynamics, Second Institute of Oceanography, MNR (No. QNHX2202).

**Data Availability Statement:** The data that support the findings of this study are available from the corresponding author Huaguo Zhang (zhanghg@sio.org.cn) upon reasonable request. The raw data used in this study are available in Google Earth Engine (https://developers.google.com/earth-engine/datasets/catalog/landsat, accessed on 11 September 2021).

**Acknowledgments:** We sincerely thank the United States Geological Survey (USGS) and Google Earth Engine for distributing the Landsat archive. The authors would like to thank Yan Li from Xiamen University for his comments on this paper.

**Conflicts of Interest:** The authors declare no conflicts of interest.

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
