# Peer review of "Mapping Annual Tidal Flat Loss and Gain in the Micro-Tidal Area Integrating Dual Full-Time Series Spectral Indices"

_remotesensing, doi:10.3390/rs16081402_

Round 1

Reviewer 1 Report

Comments and Suggestions for Authors

The main question addressed by the research is the issue of annual tidal flat loss and gain in the micro-tidal area in southern China employing remote sensing. It addresses a specific gap in the field – the limited availability of high-resolution remote sensing images with cloud-free coverage at high or low tides in wet climate zones like a humid tropical zone, but also a moderate climate zone with frequent rain and cloud cover. The research methodology is appropriate, and the presentation of the results is good, justifying the novelty of the proposed approach. I have just a few remarks to improve the quality of the manuscript.

Lines 54-57: An essential reference is necessary to support the statement since it is the underlying statement for the following paragraphs. The same is true for the statement in Lines 303-305. This statement is essential to justify the proposed tide-independent hierarchical classification strategy.

The Discussion and the Conclusions can be improved. For instance, it would be good if the authors explicitly highlight any limitations and biases of the proposed methodology in the Discussion. The conclusions could be abridged by fleshing out the main findings with global value, e.g., what could be the differences in applying the proposed methods in a moderate climate zone with frequent cloud cover?

Most paragraphs are too long. A paragraph should be 100 ± 20 words long, each dedicated to a single topic.

Comments on the Quality of English Language

I noticed a few typos and style faults throughout the text, which must be corrected.

Reviewer 2 Report

Comments and Suggestions for Authors

General Comments:

The manuscript presents a compelling methodology for mapping tidal flat changes in the micro-tidal regions, with a focus on the GHKM area. The integration of dual spectral indices and the development of an automated mapping method represent valuable contributions to the field of remote sensing. The study is well-structured, and the results are presented clearly, demonstrating the potential for practical application in coastal management and policy-making.

Specific Comments:

1.     The authors have successfully introduced a novel approach to tidal flat mapping. However, a more explicit comparison with existing methods could strengthen the manuscript. Suggest adding a brief discussion on how the proposed method outperforms or differs from traditional remote sensing techniques for clarity and context.

2. The validation process is thorough, and the accuracy metrics are commendable. It would be helpful to include a discussion on the potential impact of cloud cover and other atmospheric conditions on the accuracy of the mapping, as these factors are crucial in remote sensing applications.

3. The references are comprehensive and relevant. However, the authors might consider updating the literature review to include the most recent studies on coastal hydrodynamics.

e.g. Shi, J., Feng, X., Toumi, R., Zhang, C., Hodges, K. I., Tao, A., ... & Zheng, J. (2024). Global increase in tropical cyclone ocean surface waves. nature communications15(1), 174.

Reviewer 3 Report

Comments and Suggestions for Authors

Being not a remote sensing engineer, I focussed my review on the applicability and the reliability of the described method.

The intro is maybe what overdone, the importance of the tidal flat does not need to be elaborated in that detail. A good definition of “tidal flat” is missing. Is this the area between LLWL and HHWL or between MLW and MHW? For areas with larger tidal ranges than the GHKM area this is very relevant.

A good “ground truth” comparison with the data from your mapping is also missing in the article.

For the examples given (e.g. fig 5 and 6) it is recommended to give the exact location (coordinates), as it is done in fig 10 and 13

In fig 6  many turning points are indicated as “point”, but it seems to me a more gradual change. Natural processes of siltation and erosion seldom have a turning “point”, but human interference  has. Especially for the case in fig. 6 some more background about human interference is needed.

The paragraph on accuracy assessment (3.5) is not fully clear to me.
